# An Empirical Study on Information Extraction using Large Language Models

## Abstract

Human-like large language models (LLMs), especially the most powerful and popular ones in OpenAI's GPT family, have proven to be very helpful for many natural language processing (NLP) related tasks. Therefore, various attempts have been made to apply LLMs to information extraction (IE), which is a fundamental NLP task that involves extracting information from unstructured plain text. To demonstrate the latest representative progress in LLMs' information extraction ability, we assess the information extraction ability of GPT-4 (the latest version of GPT at the time of writing this paper) from four perspectives: Performance, Evaluation Criteria, Robustness, and Error Types. Our results suggest a visible performance gap between GPT-4 and state-of-the-art (SOTA) IE methods. To alleviate this problem, considering the LLMs' human-like characteristics, we propose and analyze the effects of a series of simple prompt-based methods, which can be generalized to other LLMs and NLP tasks. Rich experiments show our methods' effectiveness and some of their remaining issues in improving GPT-4's information extraction ability.

## 1 Introduction

The rapidly evolving field of natural language processing (NLP) witnesses the rise of large language models (LLMs), such as GPT-3(Brown et al., 2020), LaMDA(Thoppilan et al., 2022), PaLM(Chowdhery et al., 2023), etc., which have revolutionized various downstream tasks with in-context learning (ICL) (Brown et al., 2020) and chain-of-thought (COT) prompting (Wei et al., 2022c). Excitingly, just by providing appropriate instructions (Sanh et al., 2022; Ouyang et al., 2022) or chain-of-thought prompts (Wei et al., 2022c), LLMs can achieve amazing performance on the zero-shot and few-shot scenarios of unseen tasks, even without updating parameters.

Currently, one of the most popular and powerful LLM series is OpenAI's GPT series, which is best known for its two latest members, GPT-3.5 and GPT-4 (OpenAI, 2023a), by which ChatGPT is powered (OpenAI, 2023b). These two models have exhibited powerful dialogue ability and stimulated the research boom for investigating the capabilities of LLMs. For example, Jiao et al. (2023) evaluate the machine translation capability of ChatGPT, and Bang et al. (2023) assess the reasoning capability of ChatGPT. As a fundamental natural language understanding task, Information extraction (IE) aims to identify structured information of interest from unstructured plain text. Its results directly affect the subsequent downstream tasks, such as question-answering (Fei et al., 2022; Cao et al., 2022) and knowledge graph construction (Wang et al., 2022a). Besides, the LLMs' ability to recognize target information can directly reflect their performance in understanding task instructions to generate responses. This paper, therefore, aims to conduct an empirical study on information extraction using LLMs and to demonstrate the latest representative progress in LLMs' capabilities in information extraction. As the latest GPT version when writing this paper, GPT-4 has shown powerful capabilities beyond previous LLMs, including GPT-3.5. Hence, it is reasonable to select GPT-4 as a representative case of LLMs for research.

In this paper, we evaluate GPT-4's capabilities on IE tasks in terms of four perspectives, including **Performance**, **Evaluation Criteria**, **Robustness**, and **Error Types**. In response to the performance gaps

revealed in the evaluation, we also proposed three prompt-based performance **Improvement Methods** considering the human-like characteristics of GPT-4.

**Performance**    We evaluate the performance of GPT-4 on 16 datasets with 14 IE sub-tasks under 3 settings: zero-shot prompts, few-shot ICL prompts, and few-shot COT prompts. We also evaluate the performance of GPT-3.5 on these settings for more comparison, which reveals the performance improvement of the GPT series on IE tasks[1]. The results indicate the following conclusions:

- There is a significant performance gap between the two GPT models and SOTA methods. The harder the task, the larger the gap. However, the two GPT models can equal or exceed SOTA methods on a few simple tasks.

- The performance of GPT-4 is better than that of GPT-3.5 in most tasks.

- Using few-shot ICL prompts generally leads to significant improvements, but still visibly lags behind SOTA results, while the chain-of-thought prompting cannot guarantee further gains compared to few-shot ICL prompts.

**Evaluation Criteria**    Through the manual checking, we find that GPT-3.5 and GPT-4 tend to identify spans that contain or are contained by the annotated ones, i.e., the recognized spans usually contain qualifiers such as crowns, quantifiers, adjectives, time, place, etc. Thus, the previous span hard-matching strategy is not suitable for the evaluation of LLMs like GPTs that generate human-like responses. We propose a soft-matching strategy to solve this problem and display evaluation results for LLMs more accurately.

**Robustness**    We conduct comparisons and analysis on three dimensions: Invalid Output, Frequency of Target Types, and The Order of Entities. We find that:

- GPT-4 rarely outputs invalid responses in most cases.

- The frequency of target types has a significant impact on GPT-4's performance.

- GPT-4 is sensitive to the order of entities, and can understand the subject-object relationships of entities better than GPT-3.5, but still needs further improvement.

**Error Types**    We summarize 7 types of errors on IE tasks by manually checking ChatGPT's responses, including Missing spans, Missing types, Unmentioned spans, Unannotated spans, Incorrect span offsets, Undefined types, and Incorrect types. We find that "Missing spans" and "Unannotated spans" are the most dominant error types, accounting for more than 60% of errors in most cases. The widespread presence of "Unannotated spans" also raises concerns about the quality of the annotated data. Maybe using GPT-4 to assist in annotating data is a better solution.

**Improvement Methods**    As a generative large language model, GPT-4's mode in solving NLP tasks including IE is similar to that of humans. It outputs answers token by token based on the requirements specified by the instruction and input text given in the prompt, during which the rich knowledge learned from the pre-training corpus will be used. Therefore, we can consider unfine-tuned GPT-4 as a knowledgeable information extraction layman, whose IE ability can be made more proficient by designing prompts in the same way as to inspire human laymen, and this method should also apply to fine-tuned models. Based on the above analysis, we propose the following three prompt design methods:

- **Task-related Knowledge Informing**: Inform the model of the task-related knowledge required to perform the task, including The meanings of task-related terms such as "trigger" and "argument" in IE tasks.

---

[1]We use gpt-3.5-turbo-0314 and gpt-4-0125-preview for research, the latter being the latest GPT version when writing this paper.

- **Methodology Specifying**: Give a specific operation methodology to make the model more proficient in information extraction.

- **Sufficient Extraction Reminder**: When the amount of information extracted is insufficient, explicitly remind the model to extract all the required information in the text it can.

The effectiveness of these methods has been verified via rich experiments. Since the first method is the most basic one for performing tasks, it is directly applied to the performance evaluation part, and we will evaluate the performance without this method in this part for comparison.

## 2 Related Work

**Large Language Models** Based on the highly parallelizable Transformer architecture (Vaswani et al., 2017), pre-trained language models (PLMs) such as BERT (Devlin et al., 2019), BART (Lewis et al., 2020), etc., have shown powerful capabilities to solve a wide variety of NLP tasks. Some researchers find that scaling PLMs by increasing model size or data size often leads to more powerful capabilities, as long as the scaling law is followed (Kaplan et al., 2020; Hoffmann et al., 2022). Thus, numerous large-size models have been proposed, such as GPT-3 (Brown et al., 2020), LaMDA (Thoppilan et al., 2022), MT-NLG (Smith et al., 2022), PaLM (Chowdhery et al., 2023) and GPT-4 (OpenAI, 2023a), which typically have more than 100 billion parameters. The NLP community refers to these large-size PLMs as large language models (LLMs). Unlike small-sized PLMs, LLMs usually exhibit amazing emergent abilities (Wei et al., 2022b; Schaeffer et al., 2023) that enable them to achieve good performance in zero-shot and few-shot scenarios of unseen tasks, as long as the appropriate instructions (Wei et al., 2022a; Kojima et al., 2022; Wang et al., 2022b) or chain-of-though prompts (Wei et al., 2022c) are provided.

**GPT Series** One of the best-known examples of LLMs is OpenAI's GPT (Generative Pre-Training Transformer) series, including GPT-1 (Radford et al., 2018), GPT-2 (Radford et al., 2019), GPT-3 (Brown et al., 2020), GPT-4 (OpenAI, 2023a), etc. A key milestone in the development process is InstructGPT (Ouyang et al., 2022), a framework for instruction fine-tuning based on reinforcement learning from human feedback (RLHF) (Christiano et al., 2017). The framework allows a large language model to be adapted to a large number of NLP tasks simultaneously, and leverages human feedbacks to align the model output with human preferences in order to generate responses more consistent with human expectations. As the successor of InstructGPT, ChatGPT, powered by GPT-3.5 and GPT-4, has exploded the field of artificial intelligence (AI), and attracted an unprecedented wave of enthusiasm. It can interact with humans through multiple turns of dialogue, understand user intent, accomplish instructions, and return human-like responses. Shocked by ChatGPT's performance, some papers already consider GPT-4 as an early version of artificial general intelligence (AGI) (Altman, 2023; Bubeck et al., 2023).

**Information Extraction** As a popular and fundamental task, information extraction (IE) aims to extract structured knowledge of interest from unstructured plain text. The output mainly includes entities, relations between entities, event arguments, opinions, human sentiments, etc. Due to the different target information, IE mainly involves 4 tasks, including named entity recognition (NER) (Li et al., 2020; Wang et al., 2021; Ding et al., 2021; Yang et al., 2023), relation extraction (RE) (Nan et al., 2020; Zhao et al., 2021; Han et al., 2022; Zhan et al., 2022; Li et al., 2023b; Peng et al., 2022; Wang et al., 2024), event extraction (EE) (Lin et al., 2020; Lee et al., 2021a; Hsu et al., 2022) and aspect-based sentiment analysis (ABSA) (Chen & Qian, 2020; Yan et al., 2021; Feng et al., 2021; Zhang et al., 2022c;b; Yu et al., 2023). Since the result of IE directly affects the performance of subsequent higher-level applications, the importance of IE cannot be overstated. This paper intends to evaluate the performance of GPT-4 on IE, in detail.

**Evaluation of ChatGPT** ChatGPT's powerful dialog capability has triggered widespread research interest in the fields of NLP and LLM. Since ChatGPT is based on two closed models and no training details are provided, researchers are exploring its concerns and capabilities. The concerns involve ethical risks (Haque et al., 2022; Krügel et al., 2023), patient privacy (Tang et al., 2023), fabricated misinformation (Jeblick et al., 2022; Chen et al., 2023), education integrity (Malinka et al., 2023) and legal challenges (Sun, 2023). For its

capabilities, researchers evaluate the performance of ChatGPT on different tasks, including stance detection (Zhang et al., 2022a), question-answering (Guo et al., 2023), machine translation (Jiao et al., 2023), sentiment analysis (Susnjak, 2023) and other general NLP tasks (Qin et al., 2023; Zhong et al., 2023; Bian et al., 2024; Bang et al., 2023). In addition, for the information extraction task, Wei et al. (2023) propose a two-stage framework, ChatIE, to use ChatGPT for zero-shot information extraction, and evaluate its performance in detail. Li et al. (2023a) measure the performance, explainability, calibration, and faithfulness of ChatGPT on IE tasks. Following these works, this paper focuses on demonstrating the latest progress in LLMs' capabilities in information extraction represented by ChatGPT-related models. we measure the performance of the latest version of GPT-4 on multiple datasets of 14 IE subtasks, explore the impact of in-context learning (ICL) and chain-of-thought (COT) prompts on performance, evaluate robustness by scenario, and analyze error types. Following Wei et al. (2023), we also propose three novel improvement methods for GPT-4's IE performance based on its human-like characteristics. Our perspective differs significantly from Li et al. (2023a), and we evaluate more IE sub-tasks on more benchmarks.

## 3  Experimental Protocol

### 3.1  Tasks

In this paper, we consider 4 well-representative IE tasks, including Named Entity Recognition (NER), Relation Extraction (RE), Event Extraction (EE), and Aspect-based Sentiment Analysis (ABSA). Since each task contains several sub-tasks or scenarios, we conduct evaluations and analysis on the following 14 sub-tasks:

- Flat Entity Recognition (NER-Flat): Recognizing all entities within the text. Each entity is identified as a separate entity, without any hierarchical relationship between them.

- Nested Entity Recognition (NER-Nested): Recognizing all entities within the text. Each entity can be nested inside other entities, i.e., an entity may contain other sub-entities.

- Relation Classification (RE-RC): Determining the relationship between a given subject-object pair of entities within the text.

- Relational Triplet Extraction (RE-Triplet): Identifying entities and their relationships simultaneously in the form of (subject entity, relation, object entity) triplets.

- Event Detection (EE-Trigger): Identifying the word or phrase that indicates the occurrence of an event, and categorizing its corresponding event type.

- Event Argument Extraction (EE-Argument): Recognizing the entities involved in the given event, and classifying their corresponding roles.

- Trigger-Argument joint Extraction (EE-Joint): Identifying event trigger, event type, and all arguments with their roles simultaneously.

- Aspect Extraction (ABSA-AE): Extracting all the aspect terms from a review.

- Opinion Extraction (ABSA-OE): Extracting all the opinion terms from a review.

- Aspect-level Sentiment Classification (ABSA-ALSC): Predicting the sentiment polarities for every given aspect term in a review.

- Aspect-oriented Opinion Extraction (ABSA-AOE): Extracting the paired opinion terms for every given aspect term in a review.

- Aspect Extraction and Sentiment Classification (ABSA-AESC): Extracting the aspect terms as well as the corresponding sentiment polarities simultaneously.

- Pair Extraction (ABSA-Pair): Extracting the aspect terms as well as the corresponding opinion terms simultaneously.

- Triplet Extraction (ABSA-Triplet): Extracting all aspects terms with their corresponding opinion terms and sentiment polarity simultaneously.

### 3.2 Datasets

We select at least three datasets for each IE task, with a total of 16 datasets as follows:

- For NER task, the datasets include CoNLL03 (Sang & Meulder, 2003), FewNERD(Ding et al., 2021), ACE04(Doddington et al., 2004), ACE05-Ent(Walker et al., 2006), and GENIA(Ohta et al., 2002).

- For RE task, the datasets include CoNLL04 (Roth & Yih, 2004), NYT-multi (Zeng et al., 2018), and SemEval 2010 (Hendrickx et al., 2010).

- For EE task, the datasets include ACE05-Evt (Walker et al., 2006), ACE05+ (Lin et al., 2020), CASIE (Satyapanich et al., 2020), and Commodity News EE (Lee et al., 2021b).

- For ABSA task, the datasets include $D_{17}$ (Wang et al., 2017), $D_{19}$ (Fan et al., 2019), $D_{20a}$ (Peng et al., 2020), and $D_{20b}$ (Xu et al., 2020), which are all originated from the SemEval Challenges (Pontiki et al., 2014; 2015; 2016).

### 3.3 Prompts

The prompts designed in this paper all consist of five main elements: task instruction, candidate target labels, output format description, demonstration examples, and input text. The task instruction describes the specific IE sub-task, candidate target labels are the types of target information, such as entity types, relation types, etc. The output format description specifies the format of outputs to facilitate the distinguishing of target information. The demonstration examples exist under the few-shot In-context Learning setting, which can also provide the chain-of-thought explanation. The input text is a text or review from which target information is to be extracted. An example of prompts for the NER-Flat sub-task is shown in Figure 1.

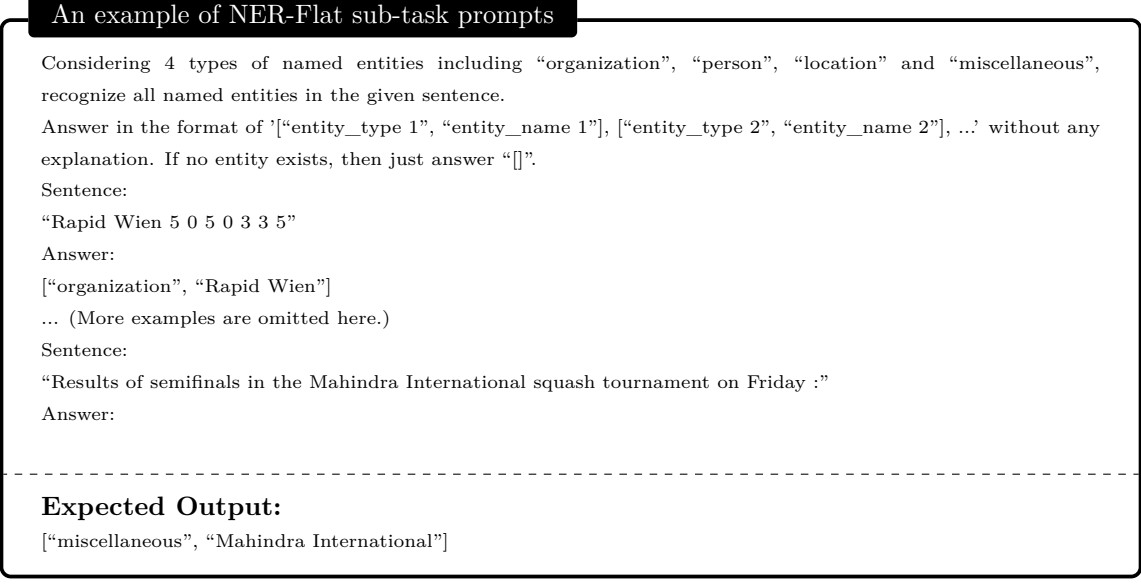

Figure 1: An example of prompts for NER-Flat sub-task on CoNLL03 dataset. See the Appendix B for more prompts.

For the demonstration examples, we randomly select them from the training set of each dataset in Section 3.2. To obtain the chain-of-thought prompts, we construct them manually with the help of ChatGPT to generate explanations.

## 3.4  Setup

To conduct a thorough evaluation of GPT-4's capabilities, for each IE sub-task, we first measure the performance of the zero-shot scenario. Then, we investigate the impact of few-shot in-context learning (ICL) and few-shot chain-of-thought (COT) prompting on the performance. For the construction of few-shot ICL prompts, we use the zero-shot prompt as the basic component and add randomly selected samples from the corresponding training set. For few-shot COT prompts, we add the chain-of-thought explanations to the few-shot ICL prompts, where the chain-of-thought explanations are manually constructed via dialog with ChatGPT. To eliminate the randomness of selected samples, we select three different groups of samples for construction and report their mean performance.

We use the official API to generate all outputs from GPT-4. To prevent the influence of dialogue history, we generate the response separately for each testing sample. Unlike other work where only 30-50 samples are selected for evaluation (Jiao et al., 2023; Wei et al., 2023), we use the entire deduplicated test set of most datasets in Section 3.2 for evaluation. Too few samples will lead to low coverage and high randomness of results, too many samples are limited by the rate and expense of accessing OpenAI's API. Since most of the datasets we use have a test set with less than 2000 samples, we limit the number of samples to a maximum of 2000 through random sampling.

Besides, we compare GPT-4, GPT-3.5, and the state-of-the-art result for each sub-task. Since calling the API of GPT-3.5 costs much less than that of GPT-4, we set its test set length limit to 3000 and select five groups of samples for few-shot ICL and COT. For the metric, we use Micro-F1 for all sub-tasks. The detailed specifications for each sub-task are as follows:

- **ABSA-AE**, **ASBA-OE**: An aspect/opinion is correct if its span matches the reference aspect/opinion mention.

- **ABSA-ALSC**: A sentiment polarity is correct if it matches the reference polarity of the given aspect term.

- **ABSA-AOE**: An opinion is correct if its span matches the reference opinion of the given aspect term.

- **ABSA-AESC**: An aspect-sentiment pair is correct if its aspect span and corresponding sentiment polarity are all correct.

- **ABSA-Pair**: An aspect-opinion pair is correct if its aspect span and opinion span all match the reference pair.

- **ABSA-Triplet**: A triplet is correct if its aspect span, opinion span, and corresponding sentiment polarity are all correct.

- **Flat-NER**, **Nested-NER**: A predicted entity is correct if its offsets and type match a reference entity.

- **RE-RC**: A predicted relation is correct if its relation type matches the reference type.

- **RE-Triplet**: A predicted relational triplet is correct if its relation type is correct and the subject and object entity spans are all correct. We only report the F1 value of relational triplets.

- **EE-Trigger**: A predicted event trigger is correct if its span and event type all match the reference trigger.

- **EE-Argument**: For a given event type, a predicted argument is correct if its span and role type all match the reference argument mention of this event.

- **EE-Joint**: A predicted argument is correct if its span, role type, and event type all match the reference argument mention. We only report the F1 value of the (event type, event argument, role) triplets contained in the output.

## 4 The Performance

In this section, we report the performance of GPT-4 and GPT-3.5 on 14 different sub-tasks, as shown in Table 1.

Table 1: The performances of GPT-4 and GPT-3.5 on different datasets over multiple standard IE tasks. "Δ F1 (%)" indicates the zero-shot performance improvement rate of GPT-4 compared to GPT-3.5, and "Ratio@SOTA" indicates the percentage value of GPT-4 performance vs. SOTA in the zero-shot scenario.

| Task | Dataset | SOTA | Zero-shot | | ICL | COT | Ratio@SOTA |
|---|---|---|---|---|---|---|---|
| | | | GPT-4 (3.5) | Δ F1 (%) | GPT-4 (3.5) | GPT-4 (3.5) | |
| ABSA-AE | $D_{17}$-14lap | 85.3 (Lv et al., 2023) | 46.06 (43.03) | 7.04% | 52.85 (48.19) | 55.92 (54.50) | 54.0% |
| | $D_{17}$-14res | 87.4 (Lv et al., 2023) | 64.74 (55.65) | 16.34% | 73.84 (70.99) | 75.71 (72.41) | 74.1% |
| | $D_{17}$-15res | 79.4 (Lv et al., 2023) | 48.71 (40.33) | 20.78% | 57.81 (53.49) | 58.37 (59.27) | 61.3% |
| ABSA-OE | $D_{17}$-14lap | 84.4 (Lv et al., 2023) | 51.44 (48.45) | 6.17% | 57.34 (57.89) | 39.26 (50.78) | 60.9% |
| | $D_{17}$-14res | 88.9 (Lv et al., 2023) | 62.68 (59.48) | 5.38% | 70.24 (71.61) | 61.92 (58.74) | 70.5% |
| | $D_{17}$-15res | 82.7 (Lv et al., 2023) | 49.49 (46.39) | 6.69% | 57.04 (53.96) | 50.77 (47.11) | 59.8% |
| ABSA-ALSC | $D_{17}$-14lap | 76.8 (Yan et al., 2021) | 79.82 (74.56) | 7.05% | 80.58 (76.76) | 79.92 (75.35) | 103.9% |
| | $D_{17}$-14res | 82.0 (Mao et al., 2021) | 86.46 (81.16) | 6.53% | 85.16 (81.85) | 87.37 (79.77) | 105.4% |
| | $D_{17}$-15res | 74.9 (Chen & Qian, 2020) | 87.06 (88.13) | -1.21% | 87.49 (86.47) | 88.60 (75.23) | 116.2% |
| ABSA-AOE | $D_{19}$-14lap | 82.2 (Feng et al., 2021) | 56.57 (57.60) | -1.78% | 62.17 (64.83) | 56.14 (55.43) | 68.8% |
| | $D_{19}$-14res | 86.4 (Feng et al., 2021) | 67.02 (67.67) | -0.96% | 71.69 (71.60) | 62.45 (66.42) | 77.6% |
| | $D_{19}$-15res | 81.6 (Feng et al., 2021) | 60.57 (67.03) | -9.64% | 64.83 (70.29) | 55.74 (59.60) | 74.2% |
| | $D_{19}$-16res | 89.2 (Feng et al., 2021) | 65.13 (73.27) | -11.10% | 71.64 (78.23) | 60.09 (68.44) | 73.0% |
| ABSA-AESC | $D_{20a}$-14lap | 70.1 (Yu et al., 2023) | 44.77 (45.48) | -1.56% | 52.70 (49.50) | 50.88 (48.87) | 63.9% |
| | $D_{20a}$-14res | 79.7 (Yu et al., 2023) | 59.94 (59.08) | 1.45% | 65.93 (65.98) | 63.69 (64.82) | 75.2% |
| | $D_{20a}$-15res | 71.6 (Yu et al., 2023) | 61.33 (53.91) | 13.77% | 65.71 (63.66) | 64.74 (66.07) | 85.7% |
| | $D_{20a}$-16res | 77.5 (Yu et al., 2023) | 56.51 (55.40) | 2.01% | 63.38 (63.11) | 65.88 (65.93) | 72.9% |
| ABSA-Pair | $D_{20a}$-14lap | 69.1 (Lv et al., 2023) | 38.20 (31.76) | 20.29% | 42.80 (41.59) | 40.91 (35.75) | 55.3% |
| | $D_{20a}$-14res | 77.8 (Zhang et al., 2021) | 48.15 (50.05) | -3.79% | 59.70 (58.88) | 55.42 (49.82) | 61.9% |
| | $D_{20a}$-15res | 69.4 (Yu et al., 2023) | 50.05 (44.41) | 12.69% | 55.52 (53.76) | 51.77 (49.62) | 72.1% |
| | $D_{20a}$-16res | 78.2 (Lv et al., 2023) | 49.91 (50.20) | -0.58% | 57.93 (58.88) | 57.52 (51.64) | 63.8% |
| ABSA-Triplet | $D_{20b}$-14lap | 61.7 (Zhang et al., 2022b) | 35.49 (33.17) | 6.99% | 40.18 (39.01) | 36.30 (33.18) | 57.5% |
| | $D_{20b}$-14res | 74.4 (Zhang et al., 2022b) | 53.20 (41.50) | 28.20% | 56.66 (54.89) | 52.66 (48.90) | 71.5% |
| | $D_{20b}$-15res | 66.1 (Zhang et al., 2022b) | 51.26 (38.89) | 31.82% | 51.31 (47.88) | 46.63 (46.55) | 77.6% |
| | $D_{20b}$-16res | 72.3 (Zhang et al., 2022b) | 54.82 (47.67) | 15.00% | 57.70 (56.55) | 54.49 (51.84) | 75.8% |
| NER-Flat | CoNLL03 | 94.6 (Wang et al., 2021) | 72.30 (60.10) | 20.30% | 78.50 (70.53) | 76.19 (74.63) | 76.4% |
| | FewNERD | 67.1 (Ding et al., 2021) | 47.84 (31.56) | 51.59% | 49.15 (36.87) | 50.61 (46.55) | 71.3% |
| NER-Nested | ACE04 | 88.5 (Yang et al., 2023) | 31.43 (27.80) | 13.05% | 43.35 (38.52) | 46.03 (40.57) | 35.5% |
| | ACE05-Ent | 87.5 (Yang et al., 2023) | 24.68 (23.38) | 5.54% | 44.68 (36.17) | 41.46 (33.98) | 28.2% |
| | GENIA | 81.5 (Yang et al., 2023) | 46.22 (38.09) | 21.36% | 56.97 (48.82) | 54.57 (50.89) | 56.7% |
| RE-RC | CoNLL04 | - | 82.07 (59.21) | 38.61% | 92.92 (55.32) | - | - |
| | NYT-multi | 93.5 (Zhan et al., 2022) | 47.79 (30.96) | 54.36% | 54.47 (26.88) | - | 51.1% |
| | SemEval2010 | 91.3 (Zhao et al., 2021) | 44.80 (39.27) | 14.09% | 44.98 (39.44) | - | 49.1% |
| RE-Triplet | CoNLL04 | 78.8 (Lou et al., 2023) | 26.53 (17.84) | 48.68% | 37.28 (24.30) | 35.88 (11.09) | 33.7% |
| | NYT-multi | 86.8 (Wang et al., 2024) | 12.74 (3.48) | 266.23% | 19.55 (12.24) | 13.41 (2.33) | 14.7% |
| | SemEval2010 | 73.2 (Yuan et al., 2023) | 5.09 (5.82) | -12.55% | 16.80 (12.85) | - | 7.0% |
| EE-Trigger | ACE05-Evt | 77.1 (Wang et al., 2023) | 41.62 (17.55) | 137.18% | 47.30 (27.33) | 27.40 (7.81) | 54.0% |
| | ACE05+ | 72.8 (Lin et al., 2020) | 43.24 (18.22) | 137.34% | 47.17 (29.17) | 26.59 (9.06) | 59.4% |
| | CASIE | 72.0 (Liu et al., 2023) | 24.98 (7.24) | 245.03% | 27.17 (18.23) | 16.92 (3.95) | 34.7% |
| | Commodity News EE | 94.0 (Lee et al., 2021a) | 47.61 (17.90) | 165.98% | 54.53 (37.79) | 27.63 (12.75) | 50.6% |
| EE-Argument | ACE05-Evt | 73.5 (Hsu et al., 2022) | 29.43 (25.09) | 17.31% | 34.40 (31.62) | - | 40.0% |
| | ACE05+ | 73.0 (Hsu et al., 2022) | 29.72 (25.80) | 15.19% | 36.32 (32.02) | - | 40.7% |
| | CASIE | - | 30.99 (17.31) | 79.01% | 31.32 (27.35) | - | - |
| | Commodity News EE | - | 16.57 (12.06) | 37.38% | 24 (15.08) | - | - |
| EE-Joint | ACE05-Evt | 57.3 (Liu et al., 2023) | 3.70 (8.74) | -57.62% | 25.52 (13.82) | - | 6.5% |
| | ACE05+ | 56.8 (Hsu et al., 2022) | 3.51 (10.12) | -65.28% | 26.19 (13.33) | - | 6.2% |
| | CASIE | 63.5 (Wang et al., 2023) | 16.78 (14.24) | 17.81% | 27.75 (18.96) | - | 26.4% |
| | Commodity News EE | 90.0 (Lee et al., 2021a) | 2.14 (8.46) | -74.67% | 20.77 (14.02) | - | 2.4% |

### 4.1 Performance Gap in Zero-shot Scenario

From the zero-shot result of Table 1, we can draw the following conclusions:

(1) **There is a significant performance gap between the two GPT models and SOTA methods.** This seems obvious and reasonable since all SOTA methods are trained on corresponding datasets. In other words, they are fully supervised models and are not zero/few-shot ones.

(2) **The harder the task, the larger the performance gap.** From the perspective of the four IE tasks of NER, RE, EE, and ABSA, it can be seen that ABSA tasks perform significantly better than RE and EE tasks. Almost all sub-tasks of ABSA for GPT-3.5 can reach more than 50% of SOTA, while all sub-tasks of RE and EE rarely exceed 30% of SOTA. For GPT-4, these two percentages are respectively 60% and 50%. One reason is that ABSA tasks involve only aspect and opinion terms and are much simpler, While RE and EE tasks involve many target types and are much harder. For example, there are 24 relation types in the NYT-multi dataset.

(3) **The harder the scenario, the larger the performance gap.** Each IE task has several scenarios. For NER tasks, the NER-Flat scenario is intuitively simpler than NER-Nested, and the performance of NER-Flat is significantly better than NER-Nested. For other tasks, including RE, EE, and ABSA, we can observe similar results.

(4) **On a few simple cases, the two GPT models can equal or exceed the performance of SOTA methods.** We can find that both models can achieve comparable performance with SOTA methods on the ABSA-ALSC sub-task, and can even surpass SOTA results, reaching up to 117.7% (GPT-3.5) and 116.2% (GPT-4) of SOTA. The sub-task is a simple sentiment classification, and the candidate polarities include "positive", "neutral" and "negative".

(5) **The performance of GPT-4 is better than GPT-3.5 in most tasks.** For 36 out of all 48 cases, the zero-shot scores of GPT-4 are higher than that of GPT-3.5. For the EE-Trigger sub-task, the improvement rates are even higher than 100%. Results prove the surprising progress achieved by the GPT series in recent times.

### 4.2 Mitigate the Gap

The observed performance gap in the above subsection is not consistent with our actual experience with ChatGPT, no matter powered by GPT-3.5 or GPT-4. To mitigate the gap, we add a few randomly selected demonstration examples to construct few-shot ICL prompts and few-shot COT prompts. We report the means of the selected example groups in Table 1.

For the few-shot ICL setting, it can be seen that "using few-shot ICL prompts generally leads to significant improvements (about 3.0∼13.0 F1 value), but still obviously lags behind SOTA results". This seems to be inconsistent with the conclusion of Wadhwa et al. (2023) that ChatGPT can achieve performance equivalent to SOTA methods by providing some demonstration examples. One reason may be that Wadhwa et al. (2023) provide more demonstration examples, i.e., almost 20 examples, while we only provide 5 demonstration examples. So, with a smaller number of demonstration examples, the few-shot ICL prompts cannot radically eliminate the performance gap.

For the few-shot COT setting, we can find that "the use of few-shot COT prompts cannot guarantee further gains compared to few-shot ICL prompts, sometimes it is worse than the performance of few-shot ICL prompts". The possible reasons are that the quality of constructed chain-of-thought prompts is not good enough and GPT-3.5 and GPT-4 are too sensitive for the few-shot COT prompts.

To sum up, we conclude that **both GPT-3.5 and GPT-4 struggle to achieve comparable performance to the corresponding SOTA methods in both zero-shot and few-shot scenarios, even if the chain-of-thought explanations are provided.**

## 5    Rethink the Gap

In this section, we rethink the performance gap from the perspective of evaluation criteria. Following the evaluation method of previous work (Lu et al., 2022; Lou et al., 2023; Liu et al., 2023), we strictly match the start and end indices of the predicted target text span (e.g., entity spans, opinion spans). This method may not be suitable for the evaluation of LLMs like GPTs that generate human-like responses.

We manually check ChatGPT's responses, and find that **GPT-3.5 and GPT-4 tend to identify spans that contain annotated spans or are contained by annotated spans, to get closer to humans.** All sub-tasks in Section 3.1 involve four types of span: entities, event triggers, aspect terms, and opinion terms. For each type of span, we select several typical annotated spans and their corresponding predicted spans and show them in Table 2. It can be seen that the annotated spans usually do not contain qualifiers such as quantifiers, articles, adjectives, time, place, etc, while the spans predicted usually contain these qualifier parts, which are also correct target information. For example, "*University of Michigan*" and "*The University of Michigan*" indicate the same target information, although the offsets are different.

Table 2: The selected annotated spans and their corresponding predicted spans.

| Type | Annotated Spans | Predicted Spans |
| --- | --- | --- |
| Entity | PGA Europro Tour
University of Michigan
Australia | 2021 PGA Europro Tour
The University of Michigan
Western Australia |
| Event Trigger | war
fighting
killed | move toward war
commit fighting forces
marines killed |
| Aspect Term | USB ports
application
cable | multiple USB ports
application crash
extender cable |
| Opinion Term | fast
well worth
not handle | super fast
well worth it
does not handle |

Therefore, to incorporate this case, we propose a soft-matching approach to obtain more accurate evaluation results, as shown in Algorithm 1, where $GetSimilarity(\cdot)$ indicates a method to calculate the similarity, for which we use the $SequenceMatcher.ratio(\cdot)$ method in the python package difflib. The similarity calculated takes the value from 0 to 1. For the threshold $\gamma$, we set it to 0.5 by default, since this process can be seen as the binary classification problem. We assume that *when the predicted span and the annotated span are only different in offset (ensured by Line 6 in the algorithm), the predicted span is reasonable and meaningful if the similarity value is higher than 0.5.*

We compare the evaluation results between the default hard-matching strategy and the soft-matching strategy for related sub-tasks and show them in Table 3. For space reasons, we only report the results of one dataset for each sub-tasks. From Table 3, it can be seen that **the soft-matching strategy delivers consistent and significant performance gains, with up to 16.50 F1 value**. Interestingly, the improvement on simple sub-tasks is much more noticeable, i.e., ABSA tasks have generally higher overall performance gains than EE tasks. Further, although the soft-matching strategy brings significant gains, it does not reach a comparable level with SOTA methods. This is still consistent with the conclusions of Section 4.

## 6    Robustness Analysis

### 6.1    Invalid Output

Since GPT-4 is a generative model, the output responses may be irrelevant information that does not meet the task requirements. In this subsection, we investigate how many invalid responses GPT-4 returns for different IE tasks. Here invalid responses refer to responses with incorrect format or unexpected content not

---

**Algorithm 1** Soft-Matching Strategy

---

**Input:** the sentence $s$, the list of annotated spans $L_A$ in sentence $s$, a predicted span $p$, the similarity threshold $\gamma$.

**Output:** Return $True$ if one of the two spans ($p$ and the annotated span $t$ with the highest similarity with $p$) contains the other and the similarity is greater than $\gamma$, otherwise return $False$.

**Begin:**

   0. $Similarity \leftarrow [\,]$

   1. **for** $t$ **in** $L_A$ :

   2.    $score \leftarrow GetSimilarity\,(t, p)$

   3.    $Similarity.append\,(score)$

   4. $score, max\_index \leftarrow max\,(Similarity)$

   5. $t \leftarrow L_A[max\_index]$

   6. **if** $p$ contains $t$ **or** $t$ contains $p$ :

   7.    **if** $score > \gamma$ :

   8.       **return** $True$.

   9. **return** $False$.

**End.**

---

Table 3: Comparison of results between default hard-matching strategy (**Hard**) and soft-matching strategy (**Soft**). $\Delta$ indicates the performance change caused by the soft-matching strategy. ABSA-ALSC and RE-RC sub-tasks are not included in this table because their required output contains no span.

| Task | Dataset | SOTA | Hard | Soft | $\Delta$F1 (%) |
|------|---------|------|------|------|------|
| ABSA-AE | $D_{17}$-14lap | 85.3 | 46.06 | 52.36 | +6.30 (13.69%) |
| ABSA-OE | $D_{17}$-14lap | 84.4 | 51.44 | 65.35 | +13.91 (27.03%) |
| ABSA-AOE | $D_{19}$-14lap | 82.2 | 56.57 | 73.07 | +16.50 (29.17%) |
| ABSA-AESC | $D_{20a}$-14lap | 70.1 | 44.77 | 52.55 | +7.79 (17.39%) |
| ABSA-Pair | $D_{20a}$-14lap | 69.1 | 38.20 | 53.7 | +15.49 (40.55%) |
| ABSA-Triplet | $D_{20b}$-14lap | 61.7 | 35.49 | 47.60 | +12.11 (34.11%) |
| NER-Flat | CoNLL03 | 94.6 | 72.30 | 74.29 | +1.99 (2.75%) |
| NER-Nested | ACE05-Ent | 87.5 | 24.68 | 33.51 | +8.83 (35.80%) |
| RE-Triplet | CoNLL04 | 78.8 | 26.53 | 34.75 | +8.22 (31.00%) |
| EE-Trigger | ACE05-Evt | 77.1 | 41.62 | 45.94 | +4.31 (10.37%) |
| EE-Argument | ACE05-Evt | 73.5 | 29.43 | 41.20 | +11.77 (40.00%) |
| EE-Joint | ACE05-Evt | 57.3 | 3.70 | 6.17 | +2.47 (66.67%) |

generated as required by task-specific prompts, which make it impossible to recognize parts of or all of the response content. The frequency and ratio of recognizable error types are analyzed in Section 7. For each sub-task, we report the ratio of invalid responses under the zero-shot setting in Table 4. For convenience, we select one dataset for each sub-task as Table 3. From the results, it can be that **in most cases, GPT-4 rarely outputs invalid responses.** However, on the EE-Joint sub-task, invalid responses account for up to 11.97%, which may result from its sophisticated format description and the difficulty of the sub-task itself.

## 6.2 Frequency of Target Types

The real-world data usually exhibits a long-tailed distribution, i.e., the frequency of target types varies greatly, causing the models to perform much worse on uncommon/tail types than on common/head ones. Here target types include entity types, relation types, event types, etc. In this subsection, we investigate the impact of the "frequency of target types" on GPT-4's performance on all IE sub-tasks. We select one dataset for each sub-task with the phenomenon of frequency differences and report the result of zero-shot prompts on the head types and tail types. For each dataset, We use a threshold $K$ to determine its head and tail types. The head types are those with more than $K$ training instances in the training set, while tail

Table 4: The ratio of invalid responses for each IE sub-task under the zero-shot setting. "#Prompt" is the number of test prompts. "#Invalid" indicates the number of test prompts with invalid responses under the zero-shot setting. "Ratio (%)" denotes the percentage of "#Invalid" in "#Prompt".

| Task | Dataset | #Prompt | #Invalid | Ratio (%) |
|------|---------|---------|----------|-----------|
| ABSA-AE | $D_{17}$-14lap | 800 | 0 | 0.00% |
| ABSA-OE | $D_{17}$-14lap | 800 | 0 | 0.00% |
| ABSA-ALSC | $D_{17}$-14lap | 654 | 0 | 0.00% |
| ABSA-AOE | $D_{19}$-14lap | 482 | 3 | 0.62% |
| ABSA-AESC | $D_{20a}$-14lap | 339 | 0 | 0.00% |
| ABSA-Pair | $D_{20a}$-14lap | 339 | 0 | 0.00% |
| ABSA-Triplet | $D_{20b}$-14lap | 328 | 0 | 0.00% |
| NER-Flat | CoNLL03 | 2000 | 0 | 0.00% |
| NER-Nested | ACE05-Ent | 1050 | 5 | 0.48% |
| RE-RC | SemEval2010 | 2000 | 4 | 0.20% |
| RE-Triplet | CoNLL04 | 287 | 1 | 0.35% |
| EE-Trigger | ACE05-Evt | 284 | 3 | 1.06% |
| EE-Argument | ACE05-Evt | 403 | 0 | 0.00% |
| EE-Joint | ACE05-Evt | 284 | 34 | 11.97% |

Table 5: The threshold (**K**), number of head types (#Head), and number of tail types (#Tail) on different datasets.

| Dataset | K | #Head | #Tail |
|---------|---|-------|-------|
| CoNLL03 | 8000 | 2 | 2 |
| ACE05-Ent | 1000 | 3 | 4 |
| SemEval2010 | 600 | 4 | 6 |
| NYT-multi | 500 | 11 | 13 |
| Commodity News EE | 100 | 7 | 12 |
| CASIE | 1200 | 2 | 3 |

Table 6: Performance comparison of GPT-4 between head and tail types for each IE sub-task. Note that the sub-tasks not listed have no long-tail distribution. "Ratio (%)" indicates the percentages of tail types' results with respect to head types' results.

| Task | Dataset | Head | Tail | Ratio (%) |
|------|---------|------|------|-----------|
| NER-Flat | CoNLL03 | 83.94 | 55.12 | 65.67% |
| NER-Nested | ACE05-Ent | 27.58 | 12.01 | 43.54% |
| RE-RC | SemEval2010 | 51.28 | 45.18 | 88.09% |
| RE-Triplet | NYT-multi | 14.50 | 2.30 | 15.89% |
| EE-Trigger | Commodity News EE | 53.01 | 37.62 | 70.97% |
| EE-Joint | CASIE | 15.73 | 17.61 | 111.89% |

types are those with less than $K$ training instances. Take the entity type "Person" as an example, if the number of "Person" entities in the training set is more than the threshold $K$, then "Person" is a head type, and vice versa, it is a tail type. The values of $K$ corresponding to each dataset used in this section are listed in Table 5. No datasets related to ABSA are shown here since ABSA tasks involve no types.

The results are shown in Table 6. It can be seen that the performance of tail types is generally significantly worse than that of head types, with the only exception of EE-Joint where their performances are close. For RE-Triplet, the performance of tail types is even lower than 16% of head types' performance. Thus, **GPT-4 also suffers from the long-tail problem**.

### 6.3 Subject-object Orders

In this subsection, we explore whether GPT-4 can distinguish the order of two entities in the RE-RC sub-task, i.e., which entity is the subject and which entity is the object. Since most relation types are not symmetric, the order of two entities is very critical. For example, the sentence "*Steven Paul Jobs was born in San Francisco on February 24, 1955.*" expresses the relational triplet ⟨*Steven Paul Jobs*, *born_in*, *San Francisco*⟩, not the triplet ⟨*San Francisco*, *born_in*, *Steven Paul Jobs*⟩, where the subject entity should take the first place while the object entity takes the last. For each instance of the asymmetric relation types, we swap the order of entities and check the change in prediction results. After exchanging the order, the prediction result should be changed to an empty string, which indicates no relation.

Table 7: Statistics of changes in predicted results after swapping the order of entities. "Changed (%)" denotes the percentage of predictions changed to empty strings when swapping subject entity and object entity. Δ means the increase of GPT-4 compared to GPT-3.5.

| Dataset | Changed (%) | | |
| --- | --- | --- | --- |
| | GPT-3.5 | GPT-4 | Δ |
| CoNLL04 | 28.11% | 84.77% | +56.66% |
| NYT-multi | 12.56% | 46.03% | +33.47% |
| SemEval2010 | 18.02% | 49.05% | +31.03% |

Table 7 shows zero-shot prompted results for this subsection. Our experiments found that GPT-4 has made large progress in identifying subject-object order over GPT-3.5, so we list the performances of both GPT-3.5 and GPT-4 in this scenario for comparison. It can be seen that only a small amount of GPT-3.5's predictions (less than 30%) can tell the difference between subject and object entities, which is denoted by the ratio of swapped predictions changed to no relation. For GPT-4, however, this ratio becomes much higher on all three datasets, with up to 84.77% on the CoNLL04 dataset, yet still not satisfying enough on the other two datasets. Therefore, it can be concluded that **for the RE-RC sub-task, GPT-4 is more sensitive to the order of entities compared to GPT-3.5, but still needs further improvement to understand their subject-object relationships accurately.**

## 7 Analysis of Error Types

In this section, we analyze GPT-4's errors on all IE sub-tasks. Here we use "span" to denote the target information to be extracted, and "types" to indicate the types of target information such as entity types, relation types, event types, sentiment polarity, etc. Through manual checking, we find that the errors mainly include:

1. **Missing spans**: Missing one or more annotated target spans.

2. **Missing types**: The annotated target span is correctly answered, but its corresponding type isn't.

3. **Unmentioned spans**: Answering the spans that do not exist within the given input text.

4. **Unannotated spans**: Answering the spans that are not annotated in the test set.

5. **Incorrect span offsets**: The offsets of the answered spans are incorrect.

6. **Undefined types**: Answering the types beyond the pre-defined types when the corresponding span is correct.

7. **Incorrect types**: The answered span is correct, while the corresponding type comes from the set of pre-defined types, but does not match the annotated type.

Here, the first two error types form the false negative samples, and error types 3 to 7 form the false positive ones. Since these error types are suitable for all sub-tasks in Section 3.1, for convenience, we take ABSA-AESC, NER-Flat, RE-Triplet, and EE-Trigger as examples (one for each task) and statistically analyze each

error type under the zero-shot setting. The ABSA tasks involve no types, so we treat the sentiment polarities of the aspects as types when analyzing ABSA-AESC. The results are shown in Table 8 and Figure 2.

Table 8: Statistical analysis of various error types for ABSA-AESC, NER-Flat, RE-Triplet, and EE-Trigger sub-tasks on $D_{20a}$-14lap, CoNLL03, CoNLL04, and ACE05-Evt datasets respectively. "Num." indicates the occurrence number of corresponding error type, while "Ratio (%)" denotes the corresponding percentage.

| Error Type | ABSA-AESC | | NER-Flat | | RE-Triplet | | EE-Trigger | |
|---|---|---|---|---|---|---|---|---|
| | Num. | Ratio (%) | Num. | Ratio (%) | Num. | Ratio (%) | Num. | Ratio (%) |
| Missing spans | 112 | 16.45% | 443 | 22.74% | 294 | 53.07% | 215 | 46.74% |
| Missing types | 102 | 14.98% | 546 | 28.03% | 27 | 4.87% | 24 | 5.22% |
| Unmentioned spans | 11 | 1.62% | 3 | 0.15% | 24 | 4.33% | 0 | 0.00% |
| Unannotated spans | 330 | 48.46% | 227 | 11.65% | 188 | 33.94% | 178 | 38.70% |
| Incorrect span offsets | 80 | 11.75% | 204 | 10.47% | 0 | 0.00% | 23 | 5.00% |
| Undefined types | 2 | 0.29% | 44 | 2.26% | 2 | 0.36% | 0 | 0.00% |
| Incorrect types | 44 | 6.46% | 481 | 24.69% | 19 | 3.43% | 20 | 4.35% |
| **Total** | 681 | 100.00% | 1948 | 100.00% | 554 | 100.00% | 460 | 100.00% |

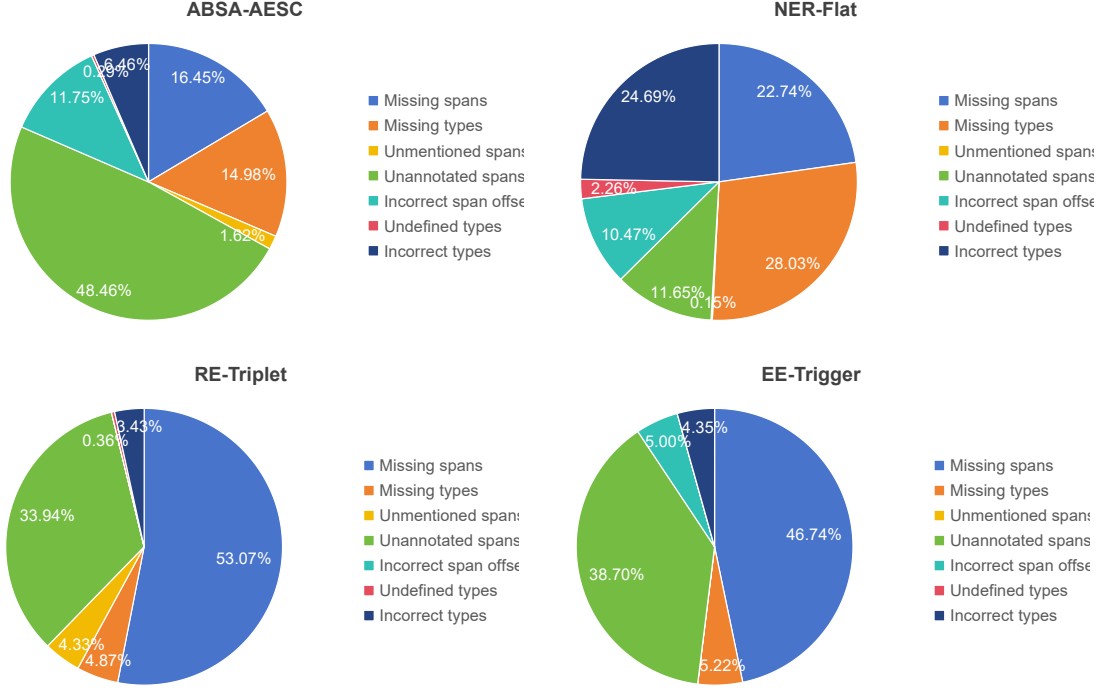

Figure 2: Percentage of error types for ABSA-AESC, NER-Flat, RE-Triplet, and EE-Trigger sub-tasks on $D_{20a}$-14lap, CoNLL03, CoNLL04, and ACE05-Evt datasets respectively.

It can be seen that **"Missing spans" and "Unannotated spans" are the two main types of errors, accounting for more than 60% in most cases.** In particular, "Unannotated spans" accounts for more than 1/3 of all errors in ABSA-AESC, RE-Triplet, and EE-Trigger sub-tasks, which raises concerns about the quality of annotated data.

## 8 Human-like Improvement Methods

In the previous sections, we conduct a comprehensive performance evaluation and robustness analysis on the information extraction ability of GPT-4. The results suggest a significant gap between the performance

of GPT-4 and the SOTA methods of IE tasks. Although GPT-4's performance has improved significantly compared to GPT-3.5, the above issues, especially its performance on difficult sub-tasks such as EE-Joint, restrict its application value in IE tasks. In this section, we propose three novel prompt-based methods for improving the IE performance of human-like LLMs like GPT-4 in the same way as to make human laymen more proficient: **Task-related Knowledge Informing**, **Methodology Specifying**, and **Sufficient Extraction Reminder**. The corresponding modified prompts are listed in B for more references.

### 8.1 Task-related Knowledge Informing

Compared with ordinary people, LLMs like GPT-3.5 and GPT-4 have the advantage of learning extensive knowledge from pre-training corpora (covering knowledge resources such as Wikipedia) and therefore have the potential to be used as knowledge bases (Tan et al., 2023). However, given the professional nature of information extraction, completing these tasks may require some task-related knowledge that is rarely seen in the corpus or easily confused with other knowledge, such as the meaning of words like "trigger" and "argument" that have a large semantic difference in the context of information extraction and other times. For a layman participating in information extraction for the first time, no matter if it is a human or an unfine-tuned LLM, informing it of the task-related knowledge in the instructions should be a basis for performing the task, without which the performance may be affected. Considering this, all prompts used in Section 4 contain the task-related knowledge required for the corresponding sub-tasks, including:

- **For NER Tasks**: Instead of giving the abbreviations of entity types, provide the model with their verbose, e.g., provide "geographical political entities" instead of "GPE".

- **For EE Tasks**: The meanings of "trigger" and "argument" in the context of EE tasks.

- **For ABSA Tasks**: The meanings of "aspect" and "opinion" in the context of ABSA tasks.

To demonstrate the effect of this prompt design method, Table 9 shows the results of the zero-shot prompt without task-related knowledge. Providing task-related knowledge does come into effect in some cases, especially in recognizing subject-object orders. Surprisingly, however, half of the presented cases witness a performance increase without task-related knowledge, especially EE-joint, where the performance increases by 13.37 F1 value. We speculate that this is because GPT-4's ability to process long texts is limited even if the text length is within its working range, which results in a decrease in its ability to understand prompts after adding additional knowledge, especially when working on complex tasks like EE-joint that require long prompts to express. In addition, GPT-4 probably can understand the type abbreviations involved in the two NER subtasks well (i.e., it may have frequent contact with them in pretraining) to correctly classify spans to these abbreviations. Therefore, it can be concluded that **providing task-related knowledge to LLMs like GPT-4 may not improve their IE performance in some cases.** The LLMs' long text processing ability and understanding of relevant knowledge should be considered when deciding whether to add it to prompts for performance improvement.

### 8.2 Methodology Specifying

In addition to informing laymen of the relevant task-related knowledge in performing IE tasks, another common way to improve performance is to specify for them a methodology to solve IE problems. In (Wei et al., 2023), the execution of IE tasks is divided into two stages, each invoked by a prompt and responsible for part of the task, and the result of the first stage paves the way for the solution of the second stage. Inspired by that, the prompt of our method is also divided into two stages with such characteristics, but we only need to send the prompt once to let the LLM solve both stages. The prompt first requires the LLM to check whether each given type of information to be extracted exists in the given text. Then, the prompt demands the LLM to output all information belonging to each existing type. Following (Wei et al., 2023), this method is not applied to ABSA tasks either, because their target information has no type.

Table 9 shows the results of the improvement method in this section on the zero-shot prompt. For most tasks, this method brings significant performance gains, with up to 16.82 F1 value on the EE-Joint sub-task. It is

Table 9: Statistical analysis of the effect of three performance improvement methods. "Base" denotes the result of the basic zero-shot prompt used in Section 4, "No TKI." indicates the result without Task-related Knowledge Informing, and "MS." and "SER." denotes the result of Methodology Specifying and Sufficient Extraction Reminder respectively. $\Delta$ denotes the performance gains and losses caused by the methods compared to "Base".

| Task | Dataset | Base | No TKI. | | MS. | | SER. | |
|---|---|---|---|---|---|---|---|---|
| | | | Result | $\Delta$ | Result | $\Delta$ | Result | $\Delta$ |
| ABSA-AE | $D_{17}$-14lap | 46.06 | 47.01 | 0.95 | - | - | 47.38 | 1.32 |
| ABSA-OE | $D_{17}$-14lap | 51.44 | 45.59 | -5.85 | - | - | 50.2 | -1.24 |
| ABSA-ALSC | $D_{17}$-14lap | 79.82 | 78.75 | -1.07 | - | - | 78.59 | -1.22 |
| ABSA-AOE | $D_{19}$-14lap | 56.57 | 52.87 | -3.7 | - | - | 57.69 | 1.11 |
| ABSA-AESC | $D_{20a}$-14lap | 44.77 | 44.63 | -0.14 | - | - | 44.73 | -0.04 |
| ABSA-Pair | $D_{20a}$-14lap | 38.2 | 41.16 | 2.96 | - | - | 39.46 | 1.26 |
| ABSA-Triplet | $D_{20b}$-14lap | 35.49 | 33.47 | -2.02 | - | - | 35.4 | -0.09 |
| NER-Flat | CoNLL03 | 72.3 | 74.38 | 2.09 | 67.25 | -5.05 | 69.23 | -3.06 |
| NER-Nested | ACE05-Ent | 24.68 | 29.96 | 5.28 | 28.98 | 4.3 | 25.81 | 1.13 |
| RE-RC | SemEval2010 | 44.80 | - | - | 43.87 | -0.93 | 44.44 | -0.37 |
| RE-Triplet | CoNLL04 | 26.53 | - | - | 28.57 | 2.05 | 24.87 | -1.65 |
| EE-Trigger | ACE05-Evt | 41.62 | 41.99 | 0.36 | 41.16 | -0.46 | 41.12 | -0.51 |
| EE-Argument | ACE05-Evt | 29.43 | 27.25 | -2.19 | 28.52 | -0.91 | 29.03 | -0.4 |
| EE-Joint | ACE05-Evt | 3.7 | 17.07 | 13.37 | 20.52 | 16.82 | 18.88 | 15.18 |

worth noting that this method generally improves more in difficult tasks, but may reduce scores on simple tasks like the NER-Flat sub-task. To conclude, **this method can be used to improve performance on difficult tasks with inspiring results.**

### 8.3 Sufficient Extraction Reminder

It can be found in Section 7 that "missing spans" is one of the major error types in GPT-4's output. Therefore, explicitly reminding the model in the prompt to extract all the required information in the text it can is a fairly obvious human-like way to improve performance. Table 9 shows the results of this method under the zero-shot setting. This method achieves a significant performance gain on EE-Joint with a 15.18 F1 value increase but has a relatively small effect on other sub-tasks. Therefore, it can be concluded that **this method is effective but narrowly adaptable and may only be suitable for very difficult tasks.**

## 9 Conclusion

In this paper, we assess the capabilities of GPT-4 from four perspectives including Performance, Evaluation Criteria, Robustness, and Error Types. We then proposed three prompt-based human-like Improvement Methods to enhance the models' performance. The details and conclusions are as follows:

**Performance** We first evaluate GPT-4's performance on 16 datasets with 14 IE sub-tasks under the zero-shot, few-shot, and chain-of-thought scenarios, and find a visible improvement over GPT-3.5 and a significant performance gap compared to SOTA results.

**Evaluation Criteria** We rethink the performance gap and find that the span hard-matching strategy is not suitable for the evaluation of GPT-4, because GPT-4 generates human-like responses. We propose a soft-matching strategy for evaluation to reflect GPT-4's performance more accurately.

**Robustness** We analyze the robustness of ChatGPT on 14 IE sub-tasks from three perspectives, including invalid output, frequency of target types, and error types. We draw the following conclusions: 1) GPT-4

rarely outputs invalid responses; 2) Long-tail target types greatly affect GPT-4's performance; 3) GPT-4 can understand the subject-object relationships in the RE-RC sub-task better than GPT-3.5, but still needs further improvement.

**Error Types**    Through manual checking, we analyze the errors of GPT-4 and summarize 7 types of errors, including Missing spans, Missing types, Unmentioned spans, Unannotated spans, Incorrect span offsets, Undefined types, and Incorrect types. We find that "Missing spans" and "Unannotated spans" are the most dominant error types. The widespread presence of "Unannotated spans" also raises concerns about the quality of previously annotated data and indicates the possibility of annotating data with GPT-4.

**Improvement Methods**    We propose three methods to improve the IE performance of LLMs like GPT-4, namely Task-related Knowledge Informing, Methodology Specifying, and Sufficient Extraction Reminder. Among them, Methodology Specifying achieves the most significant performance gains. Rich experiments prove the effectiveness of these methods, while also showing the necessity to consider the model capability, task difficulty, and characteristics when deciding whether or not to adopt these methods.

## Broader Impact Statement

In the rapidly evolving field of artificial intelligence, it's crucial to understand that models like GPT are under continuous development and refinement. The version of GPT in use today may present certain limitations or shortcomings that are an active area of research for improvement. Consequently, any tests performed or flaws discovered might only be applicable to the current iteration of the model. As time progresses and further updates are implemented, many of these initially identified issues may be addressed and corrected. Therefore, feedback and results obtained from the current usage of GPT should be contextualized as part of an ongoing development process, rather than a fixed representation of the model's performance.

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

# A    More Performance Results

We report more performances of GPT-4 for all 14 sub-tasks in Section 3.1. Under few-shot and chain-of-thought scenarios, we design three different prompts and show their mean and respective results. The results are shown in Table 10.

Table 10: More performance results.

| Task | Dataset | Zero-shot | ICL prompt 1 | ICL prompt 2 | ICL prompt 3 | ICL mean | COT prompt 1 | COT prompt 2 | COT prompt 3 | COT mean |
|------|---------|-----------|----------|----------|----------|------|----------|----------|----------|------|
| ABSA-AE | $D_{17}$-14lap | 46.06 | 55.82 | 48.66 | 54.06 | 52.85 | 55.78 | 56.95 | 55.04 | 55.92 |
|  | $D_{17}$-14res | 64.74 | 72.53 | 73.17 | 75.80 | 73.84 | 76.67 | 76.08 | 74.38 | 75.71 |
|  | $D_{17}$-15res | 48.71 | 58.71 | 60.19 | 54.53 | 57.81 | 59.02 | 58.13 | 57.96 | 58.37 |
| ABSA-OE | $D_{17}$-14lap | 51.44 | 61.72 | 53.47 | 56.82 | 57.34 | 41.92 | 34.00 | 41.86 | 39.26 |
|  | $D_{17}$-14res | 62.68 | 62.05 | 74.79 | 73.86 | 70.24 | 57.26 | 61.22 | 67.28 | 61.92 |
|  | $D_{17}$-15res | 49.49 | 57.50 | 54.88 | 58.75 | 57.04 | 45.99 | 55.06 | 51.27 | 50.77 |
| ABSA-ALSC | $D_{17}$-14lap | 79.82 | 80.28 | 79.51 | 81.96 | 80.58 | 79.05 | 80.73 | 79.97 | 79.92 |
|  | $D_{17}$-14res | 86.46 | 85.49 | 85.84 | 84.16 | 85.16 | 87.52 | 88.23 | 86.37 | 87.37 |
|  | $D_{17}$-15res | 87.06 | 87.99 | 87.43 | 87.06 | 87.49 | 89.65 | 88.17 | 87.99 | 88.60 |
| ABSA-AOE | $D_{19}$-14lap | 56.57 | 55.69 | 60.13 | 70.68 | 62.17 | 55.77 | 56.64 | 56.02 | 56.14 |
|  | $D_{19}$-14res | 67.02 | 74.08 | 67.76 | 73.22 | 71.69 | 59.38 | 62.44 | 65.53 | 62.45 |
|  | $D_{19}$-15res | 60.57 | 63.28 | 65.11 | 66.09 | 64.83 | 54.30 | 56.44 | 56.49 | 55.74 |
|  | $D_{19}$-16res | 65.13 | 70.03 | 74.32 | 70.57 | 71.64 | 60.97 | 60.21 | 59.08 | 60.09 |
| ABSA-AESC | $D_{20a}$-14lap | 44.77 | 53.44 | 51.74 | 52.92 | 52.70 | 53.92 | 50.14 | 48.59 | 50.88 |
|  | $D_{20a}$-14res | 59.94 | 65.29 | 66.13 | 66.38 | 65.93 | 65.21 | 62.23 | 63.61 | 63.69 |
|  | $D_{20a}$-15res | 61.33 | 64.59 | 65.75 | 66.80 | 65.71 | 64.75 | 64.32 | 65.14 | 64.74 |
|  | $D_{20a}$-16res | 56.51 | 61.58 | 65.48 | 63.07 | 63.38 | 66.67 | 65.39 | 65.57 | 65.88 |
| ABSA-Pair | $D_{20a}$-14lap | 38.20 | 41.83 | 42.25 | 44.31 | 42.80 | 42.35 | 39.78 | 40.61 | 40.91 |
|  | $D_{20a}$-14res | 48.15 | 58.11 | 60.40 | 60.60 | 59.70 | 52.89 | 51.90 | 61.47 | 55.42 |
|  | $D_{20a}$-15res | 50.05 | 56.62 | 54.29 | 55.66 | 55.52 | 52.70 | 50.95 | 51.66 | 51.77 |
|  | $D_{20a}$-16res | 49.91 | 54.38 | 62.08 | 57.33 | 57.93 | 57.79 | 57.51 | 57.25 | 57.52 |
| ABSA-Triplet | $D_{20b}$-14lap | 35.49 | 41.30 | 40.31 | 38.93 | 40.18 | 37.90 | 34.91 | 36.08 | 36.30 |
|  | $D_{20b}$-14res | 53.20 | 53.58 | 58.03 | 58.37 | 56.66 | 51.44 | 50.30 | 56.23 | 52.66 |
|  | $D_{20b}$-15res | 51.26 | 50.43 | 50.80 | 52.71 | 51.31 | 47.22 | 46.17 | 46.50 | 46.63 |
|  | $D_{20b}$-16res | 54.82 | 58.66 | 57.96 | 56.46 | 57.70 | 54.53 | 56.26 | 52.69 | 54.49 |
| NER-Flat | CoNLL03 | 72.30 | 77.10 | 78.91 | 79.48 | 78.50 | 76.83 | 73.93 | 77.81 | 76.19 |
|  | FewNERD | 47.84 | 50.90 | 50.41 | 46.15 | 49.15 | 52.09 | 50.05 | 49.70 | 50.61 |
| NER-Nested | ACE04 | 31.43 | 43.38 | 42.46 | 44.21 | 43.35 | 45.96 | 45.98 | 46.14 | 46.03 |
|  | ACE05-Ent | 24.68 | 44.60 | 40.51 | 48.93 | 44.68 | 41.45 | 39.12 | 43.83 | 41.46 |
|  | GENIA | 46.22 | 57.66 | 57.32 | 55.94 | 56.97 | 55.87 | 54.59 | 53.24 | 54.57 |
| RE-RC | CoNLL04 | 82.07 | 92.31 | 92.53 | 93.93 | 92.92 | - | - | - | - |
|  | NYT-multi | 47.79 | 54.33 | 55.84 | 53.24 | 54.47 | - | - | - | - |
|  | SemEval2010 | 44.80 | 44.26 | 44.61 | 46.07 | 44.98 | - | - | - | - |
| RE-Triplet | CoNLL04 | 26.53 | 35.26 | 33.05 | 43.53 | 37.28 | 34.66 | 35.14 | 37.84 | 35.88 |
|  | NYT-multi | 12.74 | 23.82 | 15.44 | 19.40 | 19.55 | 14.26 | 11.75 | 14.21 | 13.41 |
|  | SemEval2010 | 5.09 | 15.82 | 16.02 | 18.56 | 16.80 | - | - | - | - |
| EE-Trigger | ACE05-Evt | 41.62 | 50.75 | 46.61 | 44.54 | 47.30 | 25.03 | 26.93 | 30.24 | 27.40 |
|  | ACE05+ | 43.24 | 47.25 | 47.58 | 46.69 | 47.17 | 23.20 | 28.50 | 28.08 | 26.59 |
|  | CASIE | 24.98 | 33.12 | 24.33 | 24.04 | 27.17 | 19.78 | 15.38 | 15.61 | 16.92 |
|  | Commodity News EE | 47.61 | 47.58 | 59.03 | 56.98 | 54.53 | 20.47 | 33.20 | 29.23 | 27.63 |
| EE-Argument | ACE05-Evt | 29.43 | 33.27 | 33.40 | 36.55 | 34.40 | - | - | - | - |
|  | ACE05+ | 29.72 | 34.47 | 39.39 | 35.09 | 36.32 | - | - | - | - |
|  | CASIE | 30.99 | 31.62 | 31.82 | 30.52 | 31.32 | - | - | - | - |
|  | Commodity News EE | 16.57 | 24.94 | 18.48 | 28.59 | 24.00 | - | - | - | - |
| EE-Joint | ACE05-Evt | 3.70 | 24.37 | 25.37 | 26.81 | 25.52 | - | - | - | - |
|  | ACE05+ | 3.51 | 22.44 | 29.51 | 26.61 | 26.19 | - | - | - | - |
|  | CASIE | 16.78 | 28.53 | 28.83 | 25.89 | 27.75 | - | - | - | - |
|  | Commodity News EE | 2.14 | 18.49 | 17.84 | 25.99 | 20.77 | - | - | - | - |

# B    Example of input prompts

We show the zero-shot prompts, few-shot ICL prompts, few-shot COT prompts, and the prompts modified by the three improvement methods of the NER-Flat sub-task on the CoNLL03 dataset. Prompts for other datasets/tasks are similar.

**NER-Flat zero-shot prompt on the CoNLL03 dataset**

**prompt:**

Considering 4 types of named entities including "organization", "person", "location" and "miscellaneous", recognize all named entities in the given sentence.

Answer in the format of '["entity_type 1", "entity_name 1"], ["entity_type 2", "entity_name 2"], ...' without any explanation. If no entity exists, then just answer "[]".

Given sentence:

"Results of semifinals in the Mahindra International squash tournament on Friday :"

- - - - - - - - - - - - - - - - - - - - - - - - - - - - - - - - - - - - - - - - - - - - - - - - - -

**Expected Output:**

["miscellaneous", "Mahindra International"]

---

**Example NER-Flat few-shot ICL prompt on the CoNLL03 dataset**

**prompt:**

Considering 4 types of named entities including "organization", "person", "location"änd "miscellaneous", recognize all named entities in the given sentence.

Answer in the format of '["entity_type 1", "entity_name 1"], ["entity_type 2", "entity_name 2"], ...' without any explanation. If no entity exists, then just answer "[]".

Sentence:

"Rapid Wien 5 0 5 0 3 3 5"

Answer:

["organization", "Rapid Wien"]

Sentence:

"Iran accuses Iraq of ceasefire violations ."

Answer:

["location", "Iran"], ["location", "Iraq"]

... (More examples are omitted here.)

Sentence:

"The team is as follows :"

Answer:

[]

Sentence:

"Results of semifinals in the Mahindra International squash tournament on Friday :"

Answer:

- - - - - - - - - - - - - - - - - - - - - - - - - - - - - - - - - - - - - - - - - - - - - - - - - -

**Expected Output:**

["miscellaneous", "Mahindra International"]

**Example NER-Flat few-shot COT prompt on the CoNLL03 dataset**

**prompt:**

Considering 4 types of named entities including "organization", "person", "location" and "miscellaneous", recognize all named entities in the given sentence.

Answer in the format of '["entity_type 1", "entity_name 1"], ["entity_type 2", "entity_name 2"], …'. If no entity exists, then just answer "[]". Now let's think step by step. Focus your answers at the end of your response and don't print them out during your thinking. If there are multiple possible types of an entity, answer the one with the highest probability.

Sentence:

"Rapid Wien 5 0 5 0 3 3 5"

Answer:

The sentence comes from the scoreline of a soccer match. "Rapid Wien" refers to the soccer team "SK Rapid Wien", which corresponds to the "organization" in the given entity types. So, answer: ["organization", "Rapid Wien"]

Sentence:

"Iran accuses Iraq of ceasefire violations ."

Answer:

The "Iran" and "Iraq" are all countries. The country corresponds to the "location" in the given entity types. So, answer: ["location", "Iran"], ["location", "Iraq"]

… (More examples are omitted here.)

Sentence:

"The team is as follows :"

Answer:

The sentence does not involve any entity of the given entity type. So, answer: []

Your sentence:

"Results of semifinals in the Mahindra International squash tournament on Friday :"

Answer:

- - - - - - - - - - - - - - - - - - - - - - - - - - - - - - - - - - - - - - - - - - - - - - - - - - - - -

**Expected Output:**

"Mahindra International" is a sports event, which fits into "miscellaneous" since it cannot be categorized into any other given types. So, answer: ["miscellaneous", "Mahindra International"]

---

**NER-Flat zero-shot prompt with no Task-related Knowledge Informing on CoNLL03**

**prompt:**

Considering 4 types of named entities including "ORG", "PER", "LOC" and "MISC", recognize all named entities in the given sentence.

Answer in the format of '["entity_type 1", "entity_name 1"], ["entity_type 2", "entity_name 2"], …' without any explanation. If no entity exists, then just answer "[]".

Given sentence:

"Results of semifinals in the Mahindra International squash tournament on Friday :"

- - - - - - - - - - - - - - - - - - - - - - - - - - - - - - - - - - - - - - - - - - - - - - - - - - - - -

**Expected Output:**

["miscellaneous", "Mahindra International"]

---

**NER-Flat zero-shot prompt with Methodology Specifying on CoNLL03**

**prompt:**

For each of the following entity types, consider whether there are some entities of this entity type that are present in the given sentence. If present, output all entities belonging to this type in the sentence in the format of '["entity_type 1", "entity_name 1"], ["entity_type 2", "entity_name 2"], …' without any explanation, otherwise output nothing. Your answer for each existing type should take one line in response.

Given sentence:

"Results of semifinals in the Mahindra International squash tournament on Friday :¨

Given entity types: "["organization", "person", "location", "miscellaneous"]"

- - - - - - - - - - - - - - - - - - - - - - - - - - - - - - - - - - - - - - - - - - - - - - - - - - -

**Expected Output:**

["miscellaneous", "Mahindra International"]

---

**NER-Flat zero-shot prompt with Sufficient Extraction Reminder on CoNLL03**

**prompt:**

Considering 4 types of named entities including "organization", "person", "location" and "miscellaneous", recognize all named entities in the given sentence.

Answer in the format of '["entity_type 1", "entity_name 1"], ["entity_type 2", "entity_name 2"], …' without any explanation. If no entity exists, then just answer "[]". Please find all possible entities for each entity type as you can. If an entity appears multiple times in the text, answer it for the same number of times.

Given sentence:

"Results of semifinals in the Mahindra International squash tournament on Friday :"

- - - - - - - - - - - - - - - - - - - - - - - - - - - - - - - - - - - - - - - - - - - - - - - - - - -

**Expected Output:**

["miscellaneous", "Mahindra International"]

