# OpenReview forum: "An Empirical Study on Information Extraction using Large Language Models"
_TMLR — Rejected by TMLR_

### Review · Reviewer_avC9 · 2025-02-27

**Summary Of Contributions:**

This work presents a systematic comparison of GPT-4 and GPT-3.5 on information extraction capacity against SOTA methods which do not reply on large language models (LLMs). Experiments find that there remain gaps in the performance of GPT models when compared with non-LLM methods, and that GPT-4 has a better capacity when compared with GPT-3.5. This work also shows that there exists potential to improve the performance of LLM-based approach by engineering prompts.

**Audience:**

No

**Broader Impact Concerns:**

None.

**Claims And Evidence:**

No

**Requested Changes:**

- This work should investigate diverse language models, not only GPT-4 and GPT-3.5, given that the research theme is broader given the introduction. In particular, further experiments are necessary to see if the findings are also applicable to other openly accessible models, e.g., Llama.
- Further experiments are necessary to see if the capacity for information extraction is related to the instruction following abilities or knowledge capacity in large language models. I'd expect experiments by differentiating the sizes of LLMs in a consistent manner and the diverse architectures with different instruction following capabilities, in order to complement the systematic studies within this work.

**Strengths And Weaknesses:**

Strengths

- This work investigates diverse tasks related to information extraction, e.g., named entity recognition, event extraction and sentiment classification for GPT models.

Weaknesses

- The findings are not novel, but already known in several prior studies.
- Although the research theme is targeting LLMs in general, experiments are carried out only on GPT-4 and GPT-3.5. Thus, it is not clear whether the findings could be also applicable to recent models, e.g., GPT-4o, and other openly accessible models, e.g., Llama.
- This work should investigate whether the capacity for information extraction is tied with the instruction following abilities or knowledge capacity in several language models by contrastively investigating other tasks, e.g., question answering.

---

### Review · Reviewer_Nxkt · 2025-02-28

**Summary Of Contributions:**

(i) Runs a couple of existing IE benchmarks on GPT-4 and GPT-3.5, zero-shot and five-shot, and reports on the results. (ii) Uses common prompting approaches and evaluate whether that affects GPT-4 performance. (iii) Performs a more fine-grained analysis on extraction errors and properties.

**Audience:**

No

**Broader Impact Concerns:**

-

**Claims And Evidence:**

Yes

**Requested Changes:**

-

**Strengths And Weaknesses:**

S1. Summary of GPT-4 performance across various benchmarks in a certain setup

S2. Fine-grained analysis of failure cases

W1. Off topic for TMLR. This paper has no ML component and belongs to an NLP/IR venue

W2. Relevance unclear. The authors indirectly admit that their evaluation of GPT-4 is somewhat unconvincing because it's restricted to 5-shot; they state that prior work (from 2 years ago) already found 20-shot to reach much more convincing results. The authors also do not use fine-tuning (which is possible, by now even with GPT-4), a common and powerful approach in NLP. Finally, the fine-grained analysis shows problems with the benchmarks (in my point of view), which questions the entire evaluation (i.e., SOTA may have just learned idiosyncrasies of these benchmarks).

W3. Related work. There is tons of work on evaluating LLM capabilities, prompting methods, example selection for few-shot learning and so on in the IE context. This paper is largely oblivious to such works.

---

### Review · Reviewer_FHCn · 2025-03-03

**Summary Of Contributions:**

This study conducts an extensive empirical evaluation of LLM's ability to perform multiple IE tasks under various prompting strategies. It points out challenges such as long-tail entity recognition and subject-object order sensitivity. The study identifies key error types, notably missing and unannotated spans, and proposes prompt-based techniques to improve GPT-4’s IE performance. The findings indicate the potential for using LLMs in data annotation while emphasizing the need for fine-tuning and evaluation improvements in LLM-driven IE tasks.

**Audience:**

Yes

**Claims And Evidence:**

Yes

**Requested Changes:**

Refer to weakness.

**Strengths And Weaknesses:**

## Strengths
- The IE tasks and datasets included in this article are relatively comprehensive. The comparison of different model versions, as well as the proposed disadvantage of using language models for IE tasks, aligns with the observations of previous works.

- The article is generally well-written and easy to follow.

- The identification of the error types may provide some insights into the future research directory in this area.

## Weaknesses

**Critical and highly related works missing**
- The entire category of fine-tuning LLMs for IE is missing, such as [1, 2, 3, 6]. Notice that this is not a comprehensive list. Such works should not only be discussed but also be used as baselines.
- Many LLM prompting techniques for IE missing, such as [4, 5, 7]. Notice that this is not a comprehensive list. Such works should also be compared to your designed method
- Some IE data generation works such as [8] are missing. Notice that this is not a comprehensive list. Such works should be discussed in the related works.

**Overclaimed contributions**
The "entity boundary" issue has long been noticed in previous works [1, 4] and evaluation methods are adjusted accordingly in these works. The authors should not claim it as their discovery.

**Model**
In 2025 or late 2024, GPT-4 is no longer the SOTA model. Other open- and close-source models could be included to make the discussion more comprehensive.


[1] Wang, Xiao, et al. "Instructuie: Multi-task instruction tuning for unified information extraction." arXiv preprint arXiv:2304.08085 (2023).
[2] Zhou, Wenxuan, et al. "Universalner: Targeted distillation from large language models for open named entity recognition." arXiv preprint arXiv:2308.03279 (2023).
[3] Jiao, Yizhu, et al. "Instruct and extract: Instruction tuning for on-demand information extraction." arXiv preprint arXiv:2310.16040 (2023).
[4] Li, Yinghao, Rampi Ramprasad, and Chao Zhang. "A simple but effective approach to improve structured language model output for information extraction." arXiv preprint arXiv:2402.13364 (2024).
[5] Gao, Jun, et al. "Benchmarking large language models with augmented instructions for fine-grained information extraction." arXiv preprint arXiv:2310.05092 (2023).
[6] Sainz, Oscar, et al. "Gollie: Annotation guidelines improve zero-shot information-extraction." arXiv preprint arXiv:2310.03668 (2023).
[7] Averly, Reza, and Xia Ning. "Entity Decomposition with Filtering: A Zero-Shot Clinical Named Entity Recognition Framework." arXiv preprint arXiv:2407.04629 (2024).
[8] Heng, Yuzhao, et al. "ProgGen: Generating named entity recognition datasets step-by-step with self-reflexive large language models." arXiv preprint arXiv:2403.11103 (2024).

---

### Decision · Action_Editor_6QxE · 2025-04-17

**Recommendation:** Reject

**Comment:**

While the scope of the paper is relevant and reviewers appreciated the diversity of IE tasks and focused analysis of this study, the reviewers unanimously concluded that the current experimental setup of the paper is too specific and limited for publication in this venue. Authors did not submit a response throughout the discussion period. I recommend either expanding the study to allow more general conclusions or looking for a better matching venue for publishing this work and sharing the insights from this study.

**Audience:**

Some reviewers expressed doubts about relevance of the paper in its current form, but the topic is relevant and an expanded study would be relevant and of interest.

**Claims And Evidence:**

Evidence matches the focused claims on the specific model/ setup/ benchmarks in the paper, but the title/ intro make more general claims that require more extensive evaluation setup.

**Resubmission Of Major Revision:**

The authors may consider submitting a major revision at a later time.